# A Brief Overview of the Effects of Exercise and Red Beets on the Immune System in Patients with Prostate Cancer

Hadi Nobari [1,2,*], Saber Saedmocheshi [3], Kelly Johnson [4], Katsuhiko Suzuki [5,*] and Marcos Maynar-Mariño [1]

1 Department of Physiology, Faculty of Sport Sciences, University of Extremadura, 10003 Cáceres, Spain; mmaynar@unex.es
2 Department of Motor Performance, Faculty of Physical Education and Mountain Sports, Transilvania University of Braşov, 500068 Braşov, Romania
3 Department of Physical Education and Sport Sciences, Faculty of Humanities and Social Sciences, University of Kurdistan, Sanandaj 66177-15175, Iran; saedsaber384@gmail.com
4 Department of Kinesiology, Coastal Carolina University, Conway, SC 29528, USA; kjohns10@coastal.edu
5 Faculty of Sport Sciences, Waseda University, Tokorozawa 359-1192, Japan
* Correspondence: hadi.nobari1@gmail.com or nobari.hadi@unitbv.ro (H.N.); katsu.suzu@waseda.jp (K.S.)

**Abstract:** Research over the past few decades has focused on the use of functional ingredients such as an active lifestyle and proper diet as a treatment for many diseases in the world. Recent studies have shown a variety of health benefits for red beets and their active ingredients such as antioxidant, anti-inflammatory, anti-cancer, blood pressure and fat reduction, anti-diabetic, and anti-obesity effects. This review article examines the effects of exercise and red beet consumption and the effective mechanisms of these two interventions on cellular and molecular pathways in prostate cancer. However, there is a significant relationship between an active lifestyle and proper diet with the incidence of cancer, and the use of these natural interventions for cancer patients in the treatment protocol of avoidance patients. Furthermore, this review article attempts to examine the role and effect of exercise and beetroot nutrition on prostate cancer and provide evidence of the appropriate effects of using natural interventions to prevent, reduce, and even treat cancer in stages. In addition, we examine the molecular mechanisms of the effectiveness of exercise and beetroot consumption. Finally, the use of natural interventions such as exercising and eating beets due to their antioxidant, anti-inflammatory, and anti-cancer properties, due to the lack or low level of side effects, can be considered an important intervention for the prevention and treatment of cancer.

**Keywords:** anti-cancer properties; beetroot; inflammatory; obesity; physical activity

## 1. Introduction

The prostate gland is one of the most influential and pivotal organs in male reproduction under the bladder. The main function of the prostate is to create the right environment for sperm to survive during ovulation [1,2]. Cancer is more common in cells inside the prostate gland and usually occurs in older adults. The adult human prostate is divided into central, lateral, and peripheral regions, as well as fibromuscular regions and cavities [1]. Laboratory evidence suggests that stromal fibroblasts have primary carcinogenic capacity after stimulation with cancer cells, and are believed to play a role in stabilization. Epithelial cells due to high levels of androgen receptors cause hormonal dependence in prostate cancer (PCa) [1]. In addition, the secretion of prostate-specific sorantigen by these receptors increases the risk of PCa. PCa affects millions of men each year. In developed countries, PCa depends on various risk factors including age, ethnicity, genetic background, and stage of progression [3]. Histological features can show the path of disease and cancer during advanced stages. In addition to genetic factors and family history, environmental factors are also influential. Environmental factors affecting the incidence of PCa can include an active

lifestyle and proper diet. The following reasons clearly state the reason for conducting the research. These cases can reveal the study of these pathways of scientific signaling with sports and nutritional interventions for effectiveness in an overview. Due to the anatomical features of the prostate gland, most human studies have focused on their quality of life and less on the effects of exercise or dietary supplements on signals, and regarding molecular pathways, most studies have been performed on animal models [4]. A few studies have reported that an active lifestyle combined with proper nutrition can prevent, reduce, and even inhibit the progression of cancer. In the study of Saedmocheshi et al. [5] it was observed that eight weeks of aerobic exercise combined with green tea extract reduced the level of inflammatory factors in mice with PCa. Moreover, Vahabzadeh et al. [6] examined the effect of eight weeks of aerobic exercise with green tea extract on the oxidant/antioxidant balance and found that reducing the level of free radicals improved this ratio and ultimately reduced the progression of PCa. Herbert et al. [7] examined the effects of diet, physical activity, and stress reduction in patients with PCa. They found that an active lifestyle, fruit consumption, and a reduction of fatty diet control cancer. In a few studies, two or three factors affecting PCa have been studied. Due to the presence of effective compounds in red beets such as betanin and betalin, phenolics have antioxidant, anti-proliferative, and strong anti-tumor activities, while also inhibiting the growth and spread of cancer, while exercise also has the mentioned characteristics. Therefore, the article examines the interactive effect of two interventions in the prevention and treatment of cancer. Thus, there is still ambiguity in our knowledge of the effect of natural interventions on signaling pathways and proteins involved in PCs. Therefore, this review will examine the risk factors, the relationship between lifestyle and PCa, and the potential benefit of beetroot on cancer prevention. This article will first focus on the introduction of PCa and then, in order, we will discuss PCa treatment strategies, exercise, beetroot, and the simultaneous use of two interventions due to their anti-inflammatory, antioxidant, and anti-cancer properties in the treatment of PCa.

### 1.1. Prevalence and Mortality

PCa affects many men around the world every year. It is the second-most common cancer among men after lung cancer, accounting for approximately 7% of newly diagnosed cancers in men during the year [8]. Approximately 1.2 million new cases of PCa are diagnosed each year, and approximately 350,000 die each year. Aging also increases the risk of PCa and related disorders, as 85% of newly diagnosed people with PCa are over 60 years old [1,9,10].

### 1.2. Factors Involved in PCa

PCa is the result of several different factors, and comprehensive studies have suggested the role of both environmental and genetic factors in the onset and progression of the disease [11,12], some of which are presented in Table 1.

**Table 1.** Factors involved in PCa.

| Factors Involved in PCa | Mechanisms Related to Factors Leading to PCa |
| --- | --- |
| **Age** | Older men are at higher risk of PCa than others. The prevalence of this cancer increases with age compared to other cancers [13]. PCa is less common in men under the age of 40 compared to men in their 50s [14]. Its prevalence increases after the age of 50, and 75% of PCa prevalence is observed in people over 65 years of age [15]. Pervasive data suggest that the number of men dying from PCa is increasing [16]. |

**Table 1.** *Cont.*

| Factors Involved in PCa | Mechanisms Related to Factors Leading to PCa |
| --- | --- |
| **Racial origin** | Another known risk factor for PCa. According to epidemiological data, the prevalence of PCa varies between countries and ethnic differences [17]. Black people are more prone to PCa than white people, as well as Americans than Asians. In the same situation as living in the US, Ashkenazi and Icelandic Jews have a 31% higher incidence of early and more aggressive PCa, with mutations in genes such as BRCA2 [18]. |
| **Genetics** | Genetics is another important risk factor for PCa. Family history increases the risk of cancer, with approximately 9% of people with a family history having two or more relatives with PCa. Men with first-degree relatives with PCa are twice as likely to develop the disease [18]. |
| **Androgens** | Androgens are necessary for the growth, development, maintenance, and normal function of the prostate [19]. Androgen biosynthesis occurs in the testes and adrenal glands along with peripheral tissues such as the skin or prostate [20]. The two most important androgens in adult men are testosterone and its dependent metabolite, dihydrotestosterone [20]. Testosterone is the major circulating androgen that is essential for muscle mass growth, bone and cardiovascular health, sperm production regulation, and sexual function [21]. Dihydrotestosterone, on the other hand, is a functional androgen in prostate tissue and a major regulator of androgenic processes within the prostate, such as proliferation and cell differentiation [21]. In the prostate, dihydrotestosterone is made from testosterone by type 5 alpha-reductase activity [20]. Evidence supports the claim that androgens play an important role in PCa, and that high concentrations of circulating androgens are a risk factor for PCa [20]. |
| **Insulin and insulin-like growth factors** | In addition to androgens, other growth factors involved in regulating the growth of PCa cells are insulin and insulin-like growth factor-1 (IGF-1). IGF-1 is a peptide hormone involved in DNA synthesis, cell cycle stimulation, and the inhibition of cellular apoptosis [22]. In addition, it is a strong mitogen for normal and cancerous cells. Large amounts of IGF-1 are synthesized in the liver and released into the bloodstream, but some of them are produced locally within IGF-1-responsive tissues such as the prostate [23]. Thus, IGF-1 is regulated by the autocrine and paracrine mechanisms, and both mechanisms influence the action and production of IGF-1 [24]. Insulin can act as a growth factor and regulate cell differentiation, proliferation, and apoptosis [25]. The signaling pathway of insulin is similar to that of IGF-1 and its receptors; thus, these two mitogens can act similarly [26]. In the liver, insulin can stimulate IGF-1 synthesis and suppress insulin-like growth factor-binding proteins 1 and 2 (IGF-BP1 2), thus affecting the bioavailability of IGF-1 [27]. Insulin suppresses sex hormone-binding globulin (SHBG) production, which ultimately increases free testosterone levels, thus preparing the ground for PCa [28]. |
| **Obesity** | Obesity is associated with the onset and progression of several cancers, such as colon, pancreas, breast, and PCa [26,29,30]. In the US, obesity is estimated to account for 14% of all fatal cancers in men and 20% in women [31]. Figure 1 depicts the factors involved in PCa. |

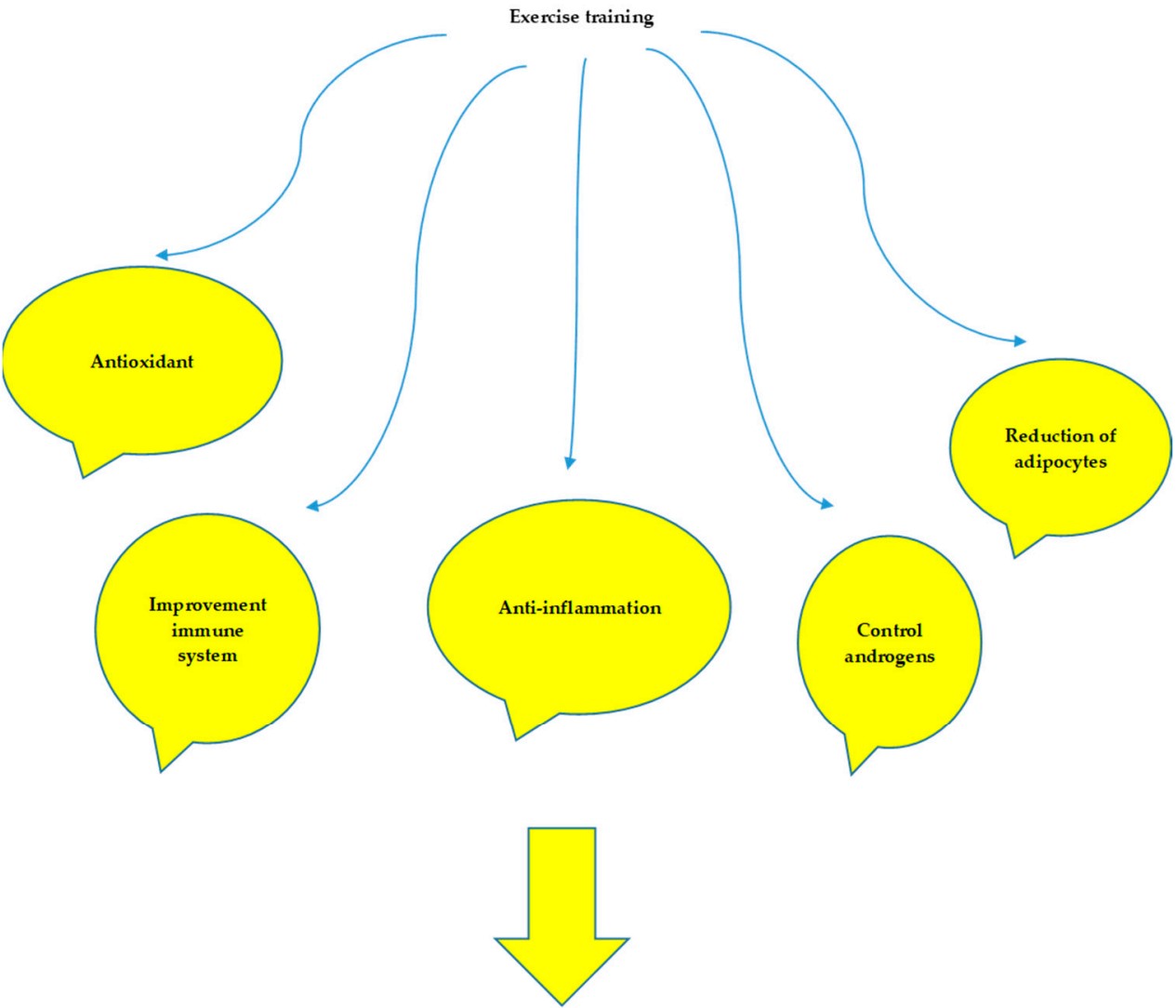

**Figure 1.** Role of exercise training in the prevention or reduction of prostate cancer.

## 2. Approach to Prevention, Control, and Treatment of PCa

Cancer patients undergoing chemotherapy may experience disorders such as fatigue, insomnia, depression, and cognitive impairment throughout their lives. Due to the long process of chemotherapy, these symptoms are present for a long time and can affect their lives. To reduce and even prevent this process, lifestyle changes can be a good solution for these patients [32]. Evidence suggests that environmental factors play a more important role in the development of the disease than genetic factors [33]. For example, low levels of physical activity and poor diet play an important role in a variety of cancers, including PCa [34,35]; it is known that physical activity potentially reduces the risk of PCa [36].

### 2.1. Exercise Training

One of the main consequences of cancer is a decrease in physical activity among people with the disease. In a way, the fear of physical activity and exercise increases among cancer patients [37].

Over the past two decades, various studies have focused on the effectiveness of exercise as a therapeutic approach to cancer treatment. Clinical trials have shown that exercise

therapy is an adjunctive tactic to improve quality of life and physical function and reduce the symptoms of cancer during treatment [38]. In addition to the role of aerobic exercise, eating habits also affect the growth and risk of cancer. Most epidemiological studies have shown that eating and aerobic exercise play an important role in the prevention of various cancers such as PCa [39]. Aerobic exercise can affect most of the pathways associated with PCa. Positive molecular mechanisms of exercise programs can be attributed to the reduction of inflammation due to low activation of nuclear factor-κB in cancer cells after exercise and growth suppression and increased apoptosis, p53, p21, and caspase activity leading to the inhibition of tumor growth indicated by tumor apoptosis and tumor suppression. Exercise can help with PCa treatment by reducing obesity and oxidative stress and modulating immune responses. Exercise reduces the level of blood circulation of testosterone and insulin-like growth factors [40].

### 2.1.1. Recommended Exercise for the Treatment or Reduction of Cancer Complications

A combination of aerobic and resistance training 3–5 times a week at a moderate intensity for 30 min is recommended for all cancer patients [41,42]. Studies show that both aerobic and resistance training improve the health and physical condition of people with cancer [43]. The benefits of moderate-intensity training have helped more cancer patients [5]. However, more research is needed to clarify the best type, intensity, and duration of exercise in the treatment of cancer. A systematic review study showed that both aerobic and resistance training models improve lower and upper torso muscle strength in cancer patients [44]. This study showed that resistance training is more effective in improving muscle strength than aerobic exercise. Among a variety of exercise models, regular exercise among people with cancer can improve physical function and reduce and even cure the disease [45]. The American College of Sports Medicine (ACSM) has recommended that both exercise models (aerobic and resistance) in cancer patients reduce fatigue, improve quality of life, improve physical function, reduce anxiety and depression, and reduce cancer-related symptoms [46]. However, ACSM recommends that aerobic exercise is an effective and safe exercise protocol for cancer treatment. Current recommendations include moderate-intensity aerobic exercise performed for 30 min at least three times per week for at least 8 to 12 weeks [46]. Furthermore, for resistance training, ACSM recommends an intensity of 60% of individuals' one-repetition maximum for at least two sets, for 8 to 15 repetitions, at least twice a week [46].

### 2.1.2. Mechanism of the Effect of Aerobic Exercise on PCa

The mechanism associated with the effect of exercise on PCa is not fully understood. Exercise over-regulates the cell cycle, and DNA regeneration pathways regulate the focal pathways involved in cell signaling, and the metabolic as well as the oxidative stress-induced nuclear factor-2 (Nrf-2) [26,47]. Studies have also shown decreased growth and increased apoptosis in androgens sensitive to human prostate cancer 2 (LNcap) cells following exercise [48]. There is evidence to suggest that exercise increases the concentration of tumor suppressor proteins such as IGFBP 1, 3, p53, and caspase, and decreases the concentration of tumor progression protein 2 expressions of B-2 cell lymphoma (BCL2) and epidermal growth factor and IGF-1 together. There is also evidence showing a direct and indirect reduction in inflammation through decreased NF-κB activity in post-exercise LNCap cells [49,50]. Aerobic exercise decreases IGFBP-3 and increases the IFG-1/IGFBP3 ratio. In the study by Mina et al., the increase in aerobic fitness was associated with the optimal concentration of leptin, the optimal ratio of leptin to adiponectin, and the ratio of IGF-1/IGFBP3 after three and six months of training [49]. Conversely, short-term to long-term moderate- to high-intensity exercise changes sex hormones and their receptors. These activities decrease the prostate-specific antigen (PSA) and the expression of androgen receptors 2 [51,52] and increase corticosterone, dihydrotestosterone, testosterone, and the expression of estrogen alpha and beta receptors [53]. Aerobic exercise also reduces inflammation by altering the C reactive protein (CRP), interleukin-6 (IL-6), tumor necrosis

factor-alpha (TNF-α), albumin, fibrinogen, and white blood cells [54]. In fact, aerobic exercise alters factors that can reduce the risk of PCa by reducing inflammation in men. Another effect of aerobic exercise is the regulation of oxidative stress. In a previous study, Guéritat et al. (2014) used four weeks of tapeworm activity in mice and had the mice perform exercise on a treadmill for 15 min, running on a treadmill at a speed of 22 m per minute for 5 days per week [55]. Their results indicated decreased prostate tumor differentiation, Bcl-2 concentration in the tumor, and increased antioxidant defense in muscles such as increased concentrations of 8-oxy-7 and 8-dihydro-2-deoxy guanosine [55]. In addition, the training protocol used in their study led to the improvement of intracellular processes, catabolic systems, autophagy-lysosomes, ubiquitin-proteasome, infection, as well as anabolic pathways such as protein synthesis. Aerobic exercise regulates tumor microenvironments by altering capillary function, hypoxia, and the oxygenation of tumor capillaries. Double exercise in mice beyond 5 to 7 weeks increases oxygen paralysis in the small vessels of the tumor and reduces hypoxia of the tumor [55]. Finally, in a series of animal studies, weeks of running on a treadmill have been shown to alter inflammatory markers of IL-6, monocyte chemoattractant protein 1 (MCP-1), and blood and apoptotic agents [56] that affect the growth and spread of PCa. All studies have shown the biological effects of aerobic exercise, while the exercise program in these studies had different frequencies, durations, and intensities. Studies on the effect of exercise on tumor growth have not shown a significant change, but the specificity of exercise in vasodilation has reduced metastasis in mice with PCa [57]. Exercise also increased apoptosis in cancer cells by increasing p53 protein levels.

Torti et al. (2004) studied the effect of exercise on PCa and expressed the mechanisms involved in the response of PCa to exercise [58]. Operational models explore how exercise can affect the risk of prostate cancer.

### 2.2. Bioactive Compounds

Combining exercise with nutrition can be a treatment modality for controlling or reducing key cancer pathways. Most studies have looked at separate improvements in diet and exercise during and after prostate cancer treatment [59]. Many studies have suggested an inverse relationship between eating vegetables and developing chronic diseases such as cancer [60,61]. These findings have shown that vegetables are rich in bioactive compounds with antioxidant capacity, which has caused more attention to them. Thus, known antioxidants in diets, such as green tea and red beets [62,63], due to their antioxidant capacity, are used as therapeutic interventions in most diseases [63]. In a review of studies on beets in cancer research, we found that the main purpose was primarily to evaluate the anti-cancer effects of beets on cancer line cells [64,65]. However, the preventive and anti-cancer effects of beet and its compounds on various mechanisms include the induction of cellular apoptosis, decreased oxidase activity, inflammatory disorders, increased anti-inflammatory cytokines, and cytotoxin function.

#### Beetroot

Beetroot is a plant with high nutrients and natural pigments. The use of beets comes from the roots themselves. Beetroot is mainly a volatile alcoholic compound due to its geosmin properties [66]. Red beets contain active compounds such as polyphenols, carotenoids, and vitamins that are useful for human health [67]. The compounds in beets have anti-cancer properties. Furthermore, beets are high in fiber, which is good for the health of the digestive system and cancer prevention [67]. The antioxidant properties of beets have led to their use in the treatment of diseases such as cardiovascular disease, anemia, impotence, and bladder stones [68]. Beets have been effective in controlling blood pressure and cardiovascular health [67,69]. Beets are also used as an ergogenic to increase energy and improve performance in athletes. In traditional Iranian medicine, beets are commonly used as a food that is effective in preventing and controlling cancer [66]. In Arabic and Chinese medicine, beetroot is used as a popular functional food for patients

with breast and colon cancer [70]. Red beets are recommended for cancer patients in many countries around the world [71]. Although observations show that red beets have a direct effect on cancer treatment due to their antioxidant role, the anti-inflammatory effects of this supplement have also been considered in certain stages of the disease [72]. Figure 2 clearly shows the beet's ability to maintain health.

- Composition of Beetroot

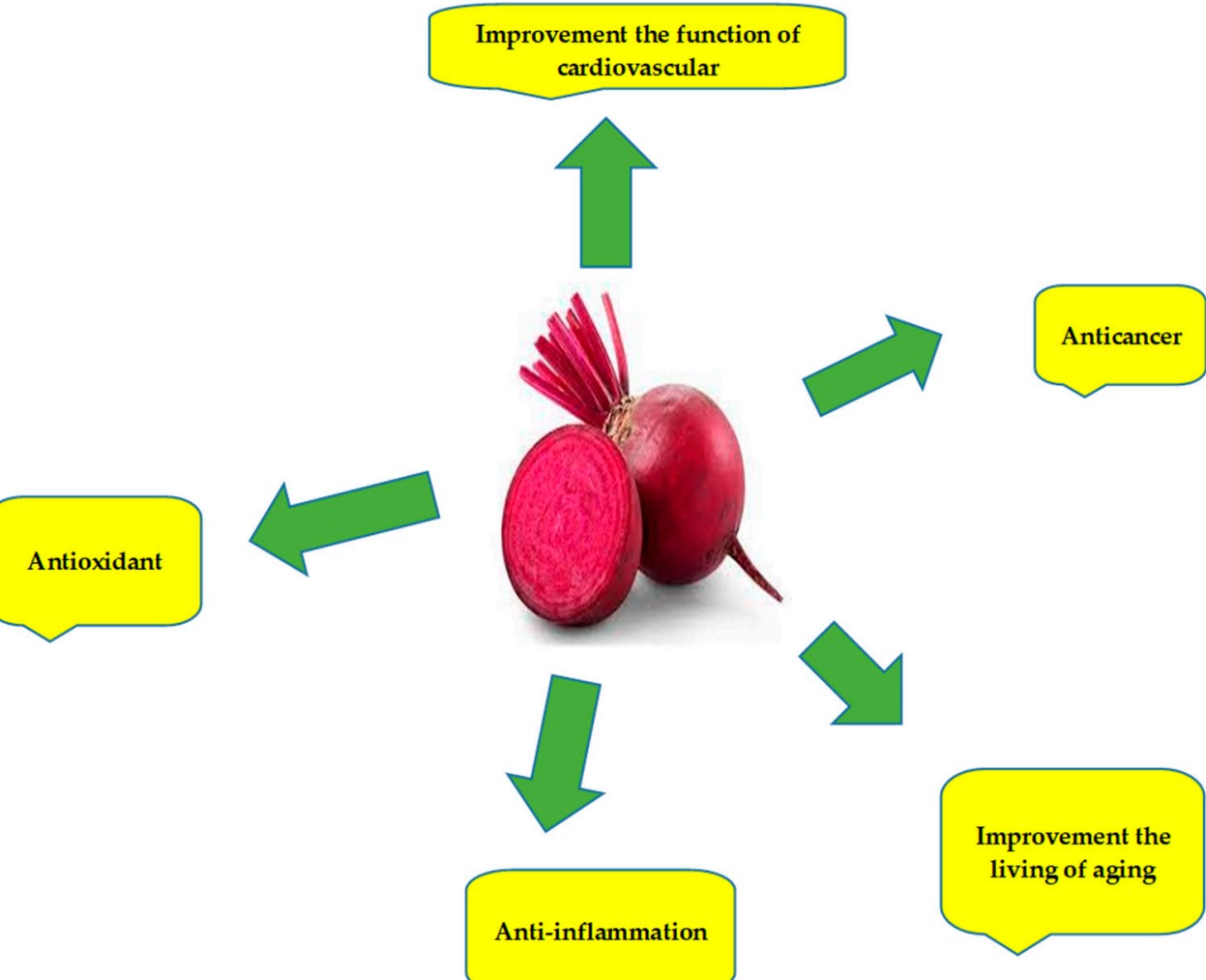

**Figure 2.** Beets' benefits for health.

The phytochemicals in beets include betalains, beta-lamic acid-derived pigments, phenolic, and flavonoids. Some betalines are also phenolic compounds, such as isobetanine, perbatanine, neobin anine, vulgaxanthin I, Vulgaxanthin II, and indicaxnthin. Other phenolic compounds in beets include 5,5,6,6-tetrahydroxy-3,3-bindolyl, *N-trans*-feruloyltyramine, *N-trans*-feruloylhomovanillylamine, and phenolic acids such as 4-hydroxybenzoic acid, chlorogenic acid, caffeic acid, Catechins, and epicatechins. In addition, flavonoids are polyphenolic metabolites that are usually beneficial to health due to their biological properties. Beet flavonoids include betagarin, beta-vulgarine, coccillin A, and dihydroisuramentin [47]. The main nutritional composition of beets includes sugar, dietary fiber, fatty acids, minerals, and vitamins. Figure 3 shows the constituent elements of beets.

- Properties of Beetroot That Have Potentially Preventive Effects on Cancer

**Figure 3.** The constituent elements of beets.

In terms of antioxidant sources, red beet is among the ten richest sources of antioxidants in Eastern and Central Europe [73]. The main elements in beets are two substances, betasianins (with the property of radical removal and antioxidants) [72] and betanine (with the properties of a phenolic group and a cyclic amine that acts as an electron donor), which act as antioxidants [72]. The consumption of beets can protect cells and tissues of the body against free radicals. Several studies have shown that beetroot is a natural antioxidant [72,74]. The antioxidant effects of beetroot are not limited to onions, as beet leaves are also rich in antioxidants [75]. Beet ethanol extract as an antioxidant, with electron donation and radical inhibition, prevented the production of nitric oxide in the macrophage of mice under RAW lipopolysaccharide 264.7 [75]. Beetroot juice contains 79.3 mg per 100 mL of betaxanthin and 159.6 mg per 100 mL of betacyanin. Taking this supplement protected cells against *N*-nitrosodiethylamine-induced oxidative stress (NDEA) and liver damage in untreated rats [76] Studies have shown that the anti-inflammatory effects of beetroot are due to its effect on the NF-κB signaling pathway [77]. The NF-κB transcription factor activates phagocytes and triggers the immune system response by activating gene targets that lead to the activation of inflammatory molecules such as chemokines and cytokines [78]. By examining the protective effect of beets against gentamicin-induced nephrotoxicity in an animal study, the expression of NF-κB, TNF-α, and IL-6, as well as myeloperoxidase activity and nitric oxide levels, were significantly reduced [78]. The consumption of beets significantly reduced the expression of caspase-3 and Bax and significantly increased the level of anti-apoptotic protein Bcl-2 in kidney cells. Studies have shown that beet consump-

tion inhibits the expression of COX-2, which is a pro-inflammatory enzyme in the synthesis of prostaglandins [76,79]. Moreover, Filippone et al. showed that beet extract caused apoptosis in breast cancer cells [80]. Beets inhibit the proliferation and death of cancer cells but do not affect normal cells. Foods containing polyphenols have anti-cancer effects. In a clinical trial study, after the intervention of polyphenolic supplementation, the level of the prostate-specific antigen (PSA) in elderly men was significantly reduced [81]. Beetroot, as food containing flavonoid polyphenolic antioxidants, can play a role in suppressing and inhibiting cancer biomarkers. In studies on the effect of beets on cell lines as an adjunct to anthracyclines as a chemotherapeutic agent [80], researchers have concluded that, because betaine (a substance in beetroot) has a similar chemical structure to Doxorubicin and has similar cytotoxic effects on cancer cells [66], it can be used as a complementary therapy. Kapadia et al. observed that red beets could reduce or inhibit the growth of prostate cancer (PC-3) and breast cancer (MCF-7) cell lines. They also found that the toxicity of red beet was significantly lower [64]. Das et al. observed that in the combination of water beet and doxorubicin in adult mice, beet reduced cardiac toxicity and cardiac cell death in cardiomyocytes [80]. Cancer cell studies have shown that beets may be an adjunct to the effective function of anthracyclines. Anthracyclines are an anti-cancer drug, but they have a side effect that leads to cardiac toxicity [68]. The researchers suggested that the presence of betanin in beets was similar to doxorubicin, which had similar cytotoxic effects on cancer cells [68], which could reduce the dose of doxorubicin and lead to cardiac toxicity.

### 3. Use of Two the Beets Exercise Interventions in Individuals with Various Diseases, including Cancer

The duration of combined diet and exercise interventions varies from 12 to 24 weeks. Each intervention should follow a specific diet based on the goal it pursues, and the diet can be planned based on the preferences of one type of diet [82]. Limited studies have been conducted to investigate the effect of two interventions of combined aerobic exercise and resistance with diets [83], and most of the subjects have been examined on one type of intervention (e.g., cycling, brisk walking, jogging, or swimming three times) for 30 min per day. Little research has examined the effects of beet consumption on improving athletic performance and health status in people with various diseases [29]. Exercise is a major strategy in modulating key signaling pathways in carcinogenesis and cancer growth. In their study, Dufrense et al. found that exercise reduced the proliferation of cancer cells. They showed that exercise reduced the growth of prostate cancer by reducing the activation of the ERK-MAPK pathway. The main mechanisms of this process are not fully understood, but several hypotheses can be proposed. Growth factors (such as EGF and IGF-1) and tyrosine kinase receptors (such as EGFR or IGF-1R) activate the ERK cascade through exercise, a process that reduces or inhibits cancer growth [84].

By reviewing studies on beets in cancer research, the focus has been on evaluating the potential chemical and anti-cancer effects of beets [85]. In summary, these effects have been mediated through various mechanisms such as the induction of cellular apoptosis, decreased oxidase activity, impaired inflammation, increased anti-inflammatory cytokines, and acting as cytotoxins (Table 2) [85]. Mancini et al., by examining the effect of red beet extract on prostate cancer cells, observed a reduction in cancer cell proliferation [65]. Kapadia et al. [64] observed a similar result by examining the effect of red beet on prostate cancer cells. Most of the studies have been related to the health effects of beet consumption on people with cardiovascular disease who have reduced aerobic capacity, intolerance to exercise, and early onset of fatigue during physical activity and daily activities [86]. Avoort et al. studied the effect of beet consumption before exercise in patients with arterial disease [87]. They showed that beets significantly delayed the onset of fatigue while walking on a treadmill. The mechanism of this process through the dilation of blood vessels and delivery of oxygen to active tissues have been mentioned [87]. Reddy et al. [88] investigated the effect of beets on cancer cell growth and observed the inhibition of MCF-7 cell growth after the consumption of beets for 48 h. The anti-cancer properties of red

beets have been investigated [66]. Supplementation of mice with 0.0025% beet before and one week after the induction of skin or liver tumors suppressed the tumor incidence [74]. Moreover, beet supplementation (40 μM) activated apoptosis [89]. By consuming red beet extract (0.299 μg/mL), an anticancer effect was observed on pancreatic (PaCa), breast (MCF-7), and prostate (PC-3) cancer cells [74]. Previous studies have shown that red beet (*Beta vulgaris* L.) has preventive and controlling activity and can reduce cell proliferation, angiogenesis, and inflammation, and can also lead to apoptosis (cd 31 and FAS) in different cell lines [74]. Many research studies have examined the relationship between exercise intensity and the inhibition of cancer cell proliferation. In animal studies, the inhibition of cancer cell proliferation was observed after moderate-intensity exercise, followed by apoptosis. Moderate-intensity training has been shown to increase Ki-67 antigen expression. Studies have shown that the effectiveness of low-intensity exercise on cancer cell proliferation is negligible, but moderate-intensity and high-intensity exercise has a high effect on cancer cell proliferation (intensity between 35% to 70% of $HR_{max}$) [90,91]. One possible mechanism in the pathway could be that moderate-intensity exercise increased dopamine levels in the brain, serum, and cancerous tissues of mouse models. By binding to dopamine receptor 2, the transfer growth factor 1 is phosphorylated and inhibits cancer cell proliferation by regulating the extracellular signal [92]. The main mechanism of this process is dependent on the hippo signal. The hippo signal pathway points to the hypothesis that phosphorylation activates homologous transcription factors, which inhibit the proliferation of cancer cells [93,94]. One of the important mechanisms by which aging increases the risk of cancer is the aging of the immune system. This process was related to reducing the function of NK cells, increasing inflammation, damaging monocytes and dendritic cells in the uptake and delivery of antigens increasing aging cells with functional failure, and reducing the number of immature T cells that respond to cancer cell development. Exercise can prevent immune aging by stimulating natural killer cells, boosting antigens, reducing inflammation, and reducing the accumulation of aging cells in the elderly [95,96]. Red beet and one of its elements, betanin, inhibits lipid membranes and peroxidation of low-density lipoprotein, modulates ROS production and gene expression to reduce the release of inflammatory cytokines, and increases the activity of antioxidant enzymes to inhibit the spread of cancer, and should therefore be part of our diet [97]. The biological effects observed after the consumption of beets in two pathways sensitive to oxidation and reduction were kappa B factor (NF-κB) and erythroid-related factor's nuclear factor—the element of antioxidant response (Nrf2-ARE) [98]. Thus, the effects of supportive therapy suggest that red beets can be included in the diet to improve the pathophysiological effects of oxidative stress and inflammatory events leading to cancer. One of the important signal pathways that play a vital role in factor signaling, angiogenesis, glycolytic metabolism, lipid metabolism, cancer cell migration, and suppression of autophagy is the mTOR pathway. Mancini et al. observed that beet leaf extract and apigenin modulated mTOR-involved proteins in both prostate cancer cell lines. Other elements in red beets are flavonoids that activate the NRF2/ARE pathway [65]. This pathway helps prevent cancer [65]. However, the activation of NRF2 in cancer cells can lead to resistance. As with the exercise-signal mechanism, the elements in beets reduce and even prevent prostate cancer due to the anti-inflammatory properties of betalin in beets by interfering with NF-κB [99].

**Table 2.** Research on the response of prostate cancer to aerobic exercise and beets.

| Author (s) | Intervention | Sample | Results |
| --- | --- | --- | --- |
| Saedmocheshi et al. [5] | Five days a week, moderate-intensity aerobic exercise for eight weeks (low- to moderate-intensity aerobic exercise training on the treadmill, 5 d·week$^{-1}$ for 45 min·d$^{-1}$ (15 min work in three sets, interspersed 2-min rest periods between sets). | Rats | Decreased inflammatory factors (NF-κB), weight loss, prostate weight loss |

**Table 2.** *Cont.*

| Author (s) | Intervention | Sample | Results |
|---|---|---|---|
| Vahabzadeh et al. [6] | Five days a week, moderate-intensity aerobic exercise for eight weeks (low- to moderate-intensity aerobic exercise training on the treadmill, 5 d·week$^{-1}$ for 45 min·d$^{-1}$ (15 min work in three sets, interspersed 2-min rest periods between sets). | Rats | Improves oxidant/antioxidant balance |
| Guéritat et al. [55] | Five days a week of moderate-intensity aerobic exercise for four weeks (5 days a week in the afternoon, one week with 15 min at 20 m/min, two weeks with 40 min at 22 m/min, and 60 min at intensity of 25 m/min for two weeks) | Rats | Decreased cancer cell differentiation in prostate tissue, increased antioxidant defense in prostate tissue |
| Mancini et al. [65] | Beetroot (betalains and flavonoids). Treatment with 100 μg/mL of beetroot extract | DU-145 and PC-3 prostate cancer cell lines | Important anti-cancer effects against prostate cancer cells |
| Kapadia et al. [100,101] | Red beetroot (*B. vulgaris* L.) extract was described as red beetroot extract diluted with Dextrin, rendered acidic with citric acid, pH 5.4, and stabilized with ascorbic acid, Batch Number GA01 with specification: λmax 530.0–536.0 nm with 1.6 min absorbance. | Human prostate cancer cells | Cytotoxicity exhibited by the red beetroot extract |

The signaling mechanisms of the effect of exercise and beet consumption on the factors involved in the incidence and progression of cancer are shown in Figure 4.

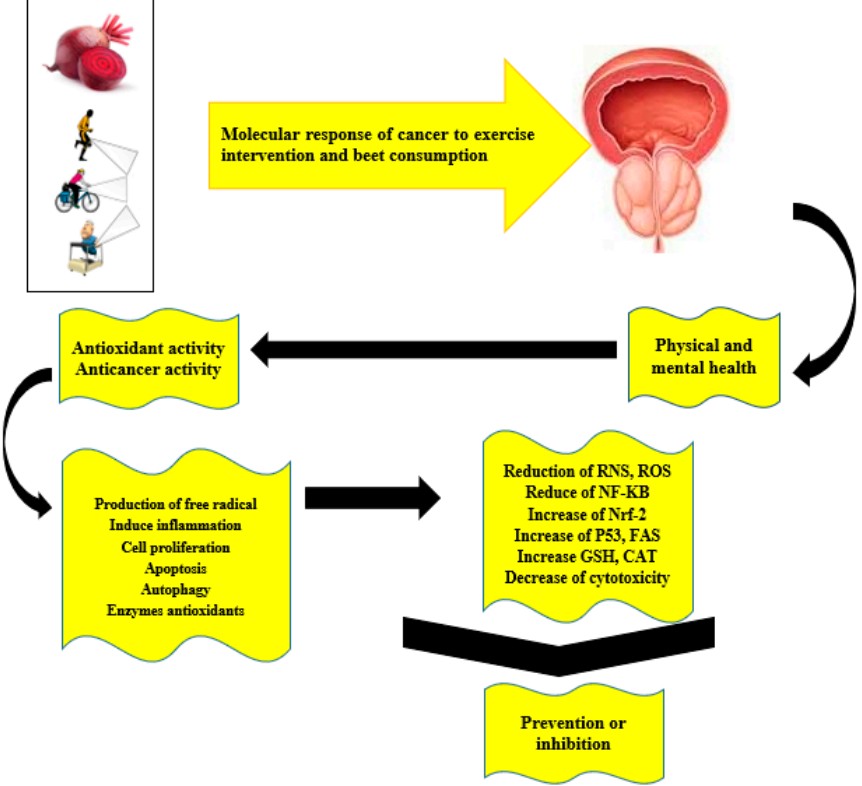

**Figure 4.** A schematic diagram of signaling mechanism effect of red beetroot and exercise training on prostate cancer.

## 4. Conclusions

To date, few studies have examined the effect of aerobic exercise and dietary supplements on PCa. One of the best adaptations of aerobic exercise is to increase antioxidant defense and strengthen the body's immune system against disease. However, its cellular-molecular underpinnings are not yet well understood. Due to the small number of studies, aerobic exercise could delay or slow the progression of cancer.

**Author Contributions:** Conceptualization, H.N., S.S. and K.J.; methodology, H.N. and S.S.; writing—original draft preparation, H.N., S.S. and K.J.; writing—review and editing, H.N., S.S., K.J., M.M.-M. and K.S. All authors have read and agreed to the published version of the manuscript.

**Funding:** This research received no external funding.

**Institutional Review Board Statement:** Not applicable.

**Informed Consent Statement:** Not applicable.

**Data Availability Statement:** Not applicable.

**Conflicts of Interest:** The authors declare no conflict of interest.

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
