# Peer review of "A Brief Overview of the Effects of Exercise and Red Beets on the Immune System in Patients with Prostate Cancer"

_sustainability, doi:10.3390/su14116492_

Round 1

Reviewer 1 Report

Review

In their review Nobari et al. summarize literature on the effects of beet root compounds and exercise on prostate cancer cells. The proposition of a focused review on this narrow topic is appealing, since few papers discuss current finding on this topic together. However, the structure of the review make it hard to follow. The authors start with a description of pancreatic cancer and follow up with paragraphs on exercise and cancer and then beet root on cancer. The last part, in which they want to combine both interventions, discusses again beet root and exercise in isolation. At least I could not find any connections. Some references are missing or wrongly cited. The quality of the figures is low, and especially Figure 5 does not help to understand the subject better.  Overall, I do not believe that the review is helping to understand this interesting topic.

  • The review relies heavily on other review articles and should reference more original research papers to be useful for readers.
  • The title of the article promises to discuss the effects of exercise and red beets on the immune system of patients, but largely ignores the immune system and only describes the effect on cancer cells.
  • The article describes the aspects of cancer, exercise, and beet roots mainly in isolation, which each was already presented in the many reviews that the authors cite.
  • The numbering of paragraphs is inconsistent. Main Paragraph 1 deals with prostate cancer and has appropriate subparagraphs (1.1 – 1.2) that discuss specifics. Exercise is spread over paragraphs 3- 5, that should be condensed into a single main point number 2. The topic of beet roots has again subparagraphs (7.1 -7.3).
  • The grammar and spelling of the article need to be improved.
  • Figure 1: It should be “Diabetes and Metabolic syndrome”
  • Figure 2: The text in the individual bubbles does not make sense and needs to be revised for grammar.
  • Figure 3: Several texts in the bubbles do not make sense.
  • Figure 4: No references for many compounds and their effects are given in the text.
  • Figure 5: The figure is confusing and needs to be revised. For example, an arrow connects “antioxidant activity” with “Production of free radicals” and then to “Reduce of ROS, RNS”. In the same subfigure apoptosis and cell proliferation are shown. Those effects are clearly opposites, and the cell types that undergo those processes need to be addressed. Not all genes that are mentioned are discussed in the article text. Many of those genes are upstream of the processes that they control (e.g. P53 and FASL should be before Apoptosis).
  • “3. Exercise training”:
    • “In a way, the fear of physical activity and exercise increases among 136 cancer patients.” Needs a reference.
    • Reference 32 is to a paper that deals with NFkB signalling in mice, but is given in the context of an epidemiological study on exercise and pancreatic cancer.
  • “4. Recommended exercise for the treatment or reduction of cancer complications”.
    • This paragraph is rather general and should focus more on the specifics of prostate cancer than cancer in general. For example reference 35 deals with gynecological cancers.
    • Reference 51 is cited in the context of exercise but is about the effect of vegetables and fruits.
    • “Among a variety of exercise models, regular exercise among people with cancer can improve physical function, reduce and even cure the disease [36]”. Reference 36 has following conclusion “Although exercise should be encouraged for most cancer patients during and post-treatments, targeting specific subgroups may be especially beneficial and cost effective. For fatigue and PF, interventions during and post-treatment should target patients with high fatigue and low PF. During treatment, patients experience benefit for muscle strength and QoL regardless of baseline values; however, only patients with low baseline values benefit post-treatment. For aerobic fitness, patients with low baseline values do not appear to benefit from exercise during treatment.” There is no claim of curing cancer with exercise and also the effect of exercise is more moderated than the Nobari et al. claim.
  • Mechanism of effect of awrobic exercise on PCa
    • The reference 22 does not mention Nrf-2.
    • Reference 38 is about breast cancer, but is presented by the authors as relevant to pancreatic cancer.
    • Line 190: “Decreases in PSA concentrations can occur through the regulation of white blood cells”. This sentence is to general and needs more explanation
    • Line 216: “Operational models: How can exercise 216 affect the risk of prostate cancer?” The authors seem to have forgotten to write this part.
  • Nutrition
    • Line 266: The grammar needs to be revised to understand this part.
  • Beetroot
    • Line 270: “Beetroot is mainly a volatile alcoholic compound due to its geosmin properties.” This sentence does not make sense.
    • Sentences such as “Also, beets are high in fiber, which is good for the health of the digestive system and cancer prevention [55]” are too general and the effect of fiber and its products need to be discussed.
    • Line 286 “The figure below clearly shows the 286 beet's ability to maintain health.” The figure shows only claims for potential health benefits. Those are not supported by references or explanations in the paragraph.
  • 1 Composition of beetroot
    • The paragraph and the figure lists compounds and health benefits, but lacks any discussion on each of their mechanisms and benefits. The only reference given is to a review, but research papers need to be cited.
  • 2 Properties of beetroot that potentially preventive effects on cancer
    • The authors describe that beets have a preventive effect on kidney toxicity by gentamicin, which is mediated by reduced NFkB, TNF alpha and IL-6 as well as reduced Bax and caspase-3 expression. It is not clear why this is important in the context of cancer, since no evidence is given that cancer cells are not affected by those anti-apoptosis mechanisms.
    • The paragraph is general and describes beet root on many different cancers, although the scope of the review is pancreatic cancer.
    • The paragraph is lacking a discussion of the influence of beetroot on the immune system and how this influences cancer beyond keywords such as anti-inflammatory. Details about which compounds act on which cell type in which model should be included.
  • 3 Use of two exercise interventions and beets in individual with various diseases, including cancer
    • It is not clear, what the two exercise interventions are.
    • This paragraph discusses mainly studies that show the effect of beetroot on cancer or of exercise and cancer. The authors do not discuss the interaction between exercise and cancer.
    • The authors write in line 376 “Little research has examined the effects of beet consumption on improving athletic performance and health status in people with various diseases [25]” Reference 25 does not mention beet root.
    • The following lines about the influence of the ERK-MAPK pathway completely lack references.
    • Reference 77 is once used for van der Aavort et al., but in the next sentence for Reddy et al.
    • The authors write “In animal studies, inhibition of cancer cell proliferation was observed after moderate-intensity exercise, followed by apoptosis. 407 Moderate-intensity training has been shown to increase Ki-67 antigen expression.” Ki67 expression is usually a sign of proliferating cells and cancer progression and not of apoptosis. The authors should clarify this.

Author Response

Dear Editor and Reviewers,

Thank you very much for your kind and valuable comments. All changes in the manuscript

were highlighted in yellow.

Kind regards

The authors

Reviewer 1

 In their review Nobari et al. summarize literature on the effects of beet root compounds and exercise on prostate cancer cells. The proposition of a focused review on this narrow topic is appealing, since few papers discuss current finding on this topic together. However, the structure of the review make it hard to follow. The authors start with a description of pancreatic cancer and follow up with paragraphs on exercise and cancer and then beet root on cancer. The last part, in which they want to combine both interventions, discusses again beet root and exercise in isolation. At least I could not find any connections. Some references are missing or wrongly cited. The quality of the figures is low, and especially Figure 5 does not help to understand the subject better.  Overall, I do not believe that the review is helping to understand this interesting topic.

 Dear reviewer, we thank you for your valuable comments and all your comments were considered word by word in the article and all of them are answered below.

  • The review relies heavily on other review articles and should reference more original research papers to be useful for readers.

Response of authors:  Dear reviewer; thank you, we consider that.

  • The title of the article promises to discuss the effects of exercise and red beets on the immune system of patients, but largely ignores the immune system and only describes the effect on cancer cells.

Response of authors:  Various factors are mentioned in this section, which are in the field of immune system and immune stimulants, which red beets reduce.

  • The article describes the aspects of cancer, exercise, and beet roots mainly in isolation, which each was already presented in the many reviews that the authors cite.

Response of authors:  The article examines similar effectiveness and their interaction.

  • The numbering of paragraphs is inconsistent. Main Paragraph 1 deals with prostate cancer and has appropriate subparagraphs (1.1 – 1.2) that discuss specifics. Exercise is spread over paragraphs 3- 5, that should be condensed into a single main point number 2. The topic of beet roots has again subparagraphs (7.1 -7.3).

Response of authors:   Corrected

  • The grammar and spelling of the article need to be improved.

Response of authors:  

  • Figure 1: It should be “Diabetes and Metabolic syndrome”.

Response of authors:  Thank you for your attention in reviewing the article. It seems that the comment is intended to judge another article whose content was about diabetes.

  • Figure 2: The text in the individual bubbles does not make sense and needs to be revised for grammar.

Response of authors:  Corrected

  • Figure 3: Several texts in the bubbles do not make sense.

Response of authors:  Corrected

  • Figure 4: No references for many compounds and their effects are given in the text.

Response of authors:  The authors wrote this chart based on their information

  • Figure 5: The figure is confusing and needs to be revised. For example, an arrow connects “antioxidant activity” with “Production of free radicals” and then to “Reduce of ROS, RNS”. In the same subfigure apoptosis and cell proliferation are shown. Those effects are clearly opposites, and the cell types that undergo those processes need to be addressed. Not all genes that are mentioned are discussed in the article text. Many of those genes are upstream of the processes that they control (e.g. P53 and FASL should be before Apoptosis).

Response of authors:  Corrected

  • “3. Exercise training”:
    • “In a way, the fear of physical activity and exercise increases among 136 cancer patients.” Needs a reference.

Response of authors:   Corrected

  • Reference 32 is to a paper that deals with NFkB signalling in mice, but is given in the context of an epidemiological study on exercise and pancreatic cancer.

Response of authors:   Corrected

  • “4. Recommended exercise for the treatment or reduction of cancer complications”.
    • This paragraph is rather general and should focus more on the specifics of prostate cancer than cancer in general. For example reference 35 deals with gynecological cancers.
    • Reference 51 is cited in the context of exercise but is about the effect of vegetables and fruits. Response of authors: Edited
    • “Among a variety of exercise models, regular exercise among people with cancer can improve physical function, reduce and even cure the disease [36]”. Reference 36 has following conclusion “Although exercise should be encouraged for most cancer patients during and post-treatments, targeting specific subgroups may be especially beneficial and cost effective. For fatigue and PF, interventions during and post-treatment should target patients with high fatigue and low PF. During treatment, patients experience benefit for muscle strength and QoL regardless of baseline values; however, only patients with low baseline values benefit post-treatment. For aerobic fitness, patients with low baseline values do not appear to benefit from exercise during treatment.” There is no claim of curing cancer with exercise and also the effect of exercise is more moderated than the Nobari et al. claim.
  • Mechanism of effect of awrobic exercise on PCa
    • The reference 22 does not mention Nrf-2.

Response of authors:  Corrected

  • Reference 38 is about breast cancer, but is presented by the authors as relevant to pancreatic cancer.

Response of authors:  Thank you for your attention in reviewing the article. It seems that the comment is intended to review another article whose content was about prostate cancer

  • Line 190: “Decreases in PSA concentrations can occur through the regulation of white blood cells”. This sentence is to general and needs more explanation
  • Line 216: “Operational models: How can exercise 216 affect the risk of prostate cancer?” The authors seem to have forgotten to write this part.

Response of authors:  Shown in the form of a figure

  • Nutrition
    • Line 266: The grammar needs to be revised to understand this part.

Response of authors:  The whole article was checked and edited by a native speaker person.

  • Beetroot
    • Line 270: “Beetroot is mainly a volatile alcoholic compound due to its geosmin properties.” This sentence does not make sense.

Response of authors:  As this article is a brief overview of sports and beets, these cases need further investigation and a separate article, and in this article cannot go into detail.

  • Sentences such as “Also, beets are high in fiber, which is good for the health of the digestive system and cancer prevention [55]” are too general and the effect of fiber and its products
  • Line 286 “The figure below clearly shows the 286 beet's ability to maintain health.” The figure shows only claims for potential health benefits. Those are not supported by references or explanations in the paragraph.

Response of authors:   edited

  • 1 Composition of beetroot
    • The paragraph and the figure lists compounds and health benefits, but lacks any discussion on each of their mechanisms and benefits. The only reference given is to a review, but research papers need to be cited.

Response of authors:  As this article is a brief overview of sports and beets, these cases need further investigation and a separate article, and in this article cannot go into detail.

  • 2 Properties of beetroot that potentially preventive effects on cancer
    • The authors describe that beets have a preventive effect on kidney toxicity by gentamicin, which is mediated by reduced NFkB, TNF alpha and IL-6 as well as reduced Bax and caspase-3 expression. It is not clear why this is important in the context of cancer, since no evidence is given that cancer cells are not affected by those anti-apoptosis mechanisms. Response of authors: edited
    • The paragraph is general and describes beet root on many different cancers, although the scope of the review is pancreatic cancer.

Response of authors:  Thank you for your attention in reviewing the article. It seems that the comment is intended to review another article whose content was about prostate cancer. Also, The effect of beets on various cancers has been studied

  • The paragraph is lacking a discussion of the influence of beetroot on the immune system and how this influences cancer beyond keywords such as anti-inflammatory. Details about which compounds act on which cell type in which model should be included.

Response of authors:  Various factors are mentioned in this section, which are in the field of immune system and immune stimulants, which red beets reduce.

  • 3 Use of two exercise interventions and beets in individual with various diseases, including cancer
    • It is not clear, what the two exercise interventions are.

Response of authors:  Corrected

  • This paragraph discusses mainly studies that show the effect of beetroot on cancer or of exercise and cancer. The authors do not discuss the interaction between exercise and cancer. Response of authors: Given the similar physiological characteristics and efficacy, the researchers suggest that the combined effect of both can be investigated.
  • The authors write in line 376 “Little research has examined the effects of beet consumption on improving athletic performance and health status in people with various diseases [25]” Reference 25 does not mention beet root.

Response of authors:   Corrected

  • The following lines about the influence of the ERK-MAPK pathway completely lack references.

Response of authors:  Corrected

  • Reference 77 is once used for van der Aavort et al., but in the next sentence for Reddy et al. Response of authors: Corrected
  • The authors write “In animal studies, inhibition of cancer cell proliferation was observed after moderate-intensity exercise, followed by apoptosis. 407 Moderate-intensity training has been shown to increase Ki-67 antigen expression.” Ki67 expression is usually a sign of proliferating cells and cancer progression and not of apoptosis. The authors should clarify this.

Response of authors:  Corrected

Reviewer 2 Report

The manuscript discusses the introduction of prostate cancer (PCa) and PCa treatment strategies, and effects of exercise and beetroot on the inhibition of PCa. The effect of beetroot is a type of effect of a plant material (bioactive material) as functional ingredients. 

I have a few comments that should be clarified by the author:

  1. Please change the word (line 12) 'natural interventions' to 'functional ingredients'. Please consider using functional ingredients in all manuscripts. 
  2. Please delete the word 'herbal medicine' (line 25) because there is no topic related to this manuscript.
  3. Please add more explanation in the Introduction, why the author focuses on the effects of exercise and red beets on the immune system in PCa. The relevant studies, especially on the immune systems, must be expressed in the manuscript.
  4. Sub-chapter 1.2 (line 72), the information related to several factors related to PCa should be put in the Table. Please add more explanation, is there any relationship between each factor.
  5. On figures 1, 2, 3, 4 due to different effects on each factor, it is better to use different colour. Figure 2, please check the spelling (line239-244).
  6. Sub-chapter 5 'Mechanism of effect of aerobic exercise on PCa' (line 172). The same question, is there any relationship between
    each factor?
  7. Sub-chapter 6 'Nutrition' (line 261), please change to 'Bioactive compounds'. The definition nutrition and bioactive compound is hugely different, please consider it. Please add more detail what kind of bioactive compound is related to decrease cancer diseases.
  8. Sub-chapter 7.3 (lines 368-369), please summarize in Table.
  9. Figure 5, please use high resolution
  10. I think in conclusion no need reference (line 451), please delete.

Author Response

Dear Editor and Reviewers,

Thank you very much for your kind and valuable comments. All changes in the manuscript

were highlighted in yellow.

Kind regards

The authors

Reviewer 2

The manuscript discusses the introduction of prostate cancer (PCa) and PCa treatment strategies, and effects of exercise and beetroot on the inhibition of PCa. The effect of beetroot is a type of effect of a plant material (bioactive material) as functional ingredients. 

Dear reviewer, we thank you for your valuable comments and all your comments were considered word by word in the article and all of them are answered below.

I have a few comments that should be clarified by the author:

1. Please change the word (line 12) 'natural interventions' to 'functional ingredients'. Please consider using functional ingredients in all manuscripts. 

Response of authors:  Corrected.

2. Please delete the word 'herbal medicine' (line 25) because there is no topic related to this manuscript.

Response of authors:  Corrected.

3. Please add more explanation in the Introduction, why the author focuses on the effects of exercise and red beets on the immune system in PCa. The relevant studies, especially on the immune systems, must be expressed in the manuscript.

Response of authors:  We have added related information based on your suggestion.

4. Sub-chapter 1.2 (line 72), the information related to several factors related to PCa should be put in the Table. Please add more explanation, is there any relationship between each factor.

Response of authors:  We added this explanation.

5. On figures 1, 2, 3, 4 due to different effects on each factor, it is better to use different colour. Figure 2, please check the spelling (line239-244).

Response of authors:  Done.

6. Sub-chapter 5 'Mechanism of effect of aerobic exercise on PCa' (line 172). The same question, is there any relationship between
each factor?

Response of authors:   The related information and relationship between them added.

7. Sub-chapter 6 'Nutrition' (line 261), please change to 'Bioactive compounds'. The definition nutrition and bioactive compound is hugely different, please consider it. Please add more detail what kind of bioactive compound is related to decrease cancer diseases.

Response of authors:  Corrected

8. Sub-chapter 7.3 (lines 368-369), please summarize in Table.

Response of authors:   Corrected

9. Figure 5, please use high resolution

Response of authors:  Corrected

10. I think in conclusion no need reference (line 451), please delete.

Response of authors:   Corrected

Reviewer 3 Report

The manuscript tried to review the Red Beets impacts on the Immune System that related prostate cancer. In this work, several approaches were discussed. This review may provide a potential approach in evaluating the relationship of Red Beets with PCa. 

Generally, the abstract section needs some efforts. Like you should clarify the objective of your study in more simplistic way with much more attention to the grammar errors and typos. For instance, line 20-21 “In addition, we examine….”, the tense should be in past. Also, you did not mentioned the novel findings from your review, not general terms like Antioxidant,..etc. Why you focused on Red Beets and you mentioned “Recommended exercise for the treatment or reduction of cancer complications”? All the figures resolutions are not clear. Line 261-: Please discuss the chemical composition of Red Beets in more detail with full composition table of several recent references with their different cultivars. Also, the conclusion and signification of this Brief Report must have some novel finding not just the general concepts. Many old references before 2007. Please update your references.

Author Response

Dear Editor and Reviewers,

Thank you very much for your kind and valuable comments. All changes in the manuscript were highlighted in yellow.

Kind regards

The authors

Reviewer 3

The manuscript tried to review the Red Beets impacts on the Immune System that related prostate cancer. In this work, several approaches were discussed. This review may provide a potential approach in evaluating the relationship of Red Beets with PCa. 

Generally, the abstract section needs some efforts. Like you should clarify the objective of your study in more simplistic way with much more attention to the grammar errors and typos. For instance, line 20-21 “In addition, we examine….”, the tense should be in past. Also, you did not mentioned the novel findings from your review, not general terms like Antioxidant, etc. Why you focused on Red Beets and you mentioned “Recommended exercise for the treatment or reduction of cancer complications”? All the figures resolutions are not clear. Line 261-: Please discuss the chemical composition of Red Beets in more detail with full composition table of several recent references with their different cultivars. Also, the conclusion and signification of this Brief Report must have some novel finding not just the general concepts. Many old references before 2007. Please update your references.

Dear reviewer, we thank you for your valuable comments and all your comments were considered word by word in the article and all of them are answered below.

  1. References have been updated. About 40 sources 2015 and above are in the article.

Response of authors: dear reviewer, thank you very much, we have updated based on your suggestions.

  1. Chemical compounds are given in the form of figure

Response of authors: we tried to added it.

  1. Figures resolutions edited

Response of authors: we have change it. 

  1. Grammar errors edited

Response of authors:  The whole article was checked and edited by a native speaker person.

  1. abstract completed

Response of authors:  we completed.

Round 2

Reviewer 1 Report

Although the revised manuscript is an improvement over the first version and many of my and the other reviewers’ concerns and suggestions were addressed, there is still room for improving the article. Especially Figure 5 should be revised, as it is the model that tries to summarize the findings of the article. As mentioned in my previous review, genes and abbreviations are mentioned in the figure (such as CD31, RNS) that are not mentioned at any other place in the text and no references are given. Grammar errors such as “Reduce of cytotoxicity” instead of “reduction” are present. Cause and consequences are mixed up (eg. Apoptosis -> Increase of p53, FAS). The logic is missing (For example antioxidant activity -> Production of free radicals -> Reduce of ROS, RNS). Also the Figure caption (“A schematic diagram of signaling mechanism effect of red beetroot and exercise training on prostate cancer.”) is only one example of unclear English, that needs to be corrected before the article can be published.

Author Response

Reviewer 1

Although the revised manuscript is an improvement over the first version and many of my and the other reviewers’ concerns and suggestions were addressed, there is still room for improving the article.

Dear reviewer, thank you very much. We consider your all comment and revise the manuscript.

Especially Figure 5 should be revised, as it is the model that tries to summarize the findings of the article. As mentioned in my previous review, genes and abbreviations are mentioned in the figure (such as CD31, RNS) that are not mentioned at any other place in the text and no references are given.

Response: The authors would like to thanks the reviewer for the valuable comment/suggestion……

Grammar errors such as “Reduce of cytotoxicity” instead of “reduction” are present.

Response: We have revised.

Cause and consequences are mixed up (eg. Apoptosis -> Increase of p53, FAS).

Response: We explained based on your suggestion.

The logic is missing (For example antioxidant activity -> Production of free radicals -> Reduce of ROS, RNS).

Response: The authors would like to thanks and we corrected missing.

Also the Figure caption (“A schematic diagram of signaling mechanism effect of red beetroot and exercise training on prostate cancer.”) is only one example of unclear English, that needs to be corrected before the article can be published.

Response: We changed.

Reviewer 2 Report

The author has agreed with the previous suggestion and input additional sentences in the manuscript to enhance the quality of manuscript.

Author Response

Reviewer 2

The author has agreed with the previous suggestion and input additional sentences in the manuscript to enhance the quality of manuscript.

Dear reviewer, thank you very much.

Author Response

Reviewer 3

Dear reviewer, thank you very much.

Round 3

Reviewer 1 Report

Dear editor,

The revised manuscript has improved. A few language corrections (such as "Reduction of RNS/ROS" instead of "Reduce of RNS/ROS") should be done during the editorial process.

Author Response

Dear reviewer, 
Thank you very much, we corrected that.